# Bioinformatic Analysis of 1000 Amphibian Antimicrobial Peptides Uncovers Multiple Length-Dependent Correlations for Peptide Design and Prediction

**DOI:** 10.3390/antibiotics9080491

**Published:** 2020-08-07

**Authors:** Guangshun Wang

**Affiliations:** Department of Pathology and Microbiology, College of Medicine, University of Nebraska Medical Center, 985900 Nebraska Medical Center, Omaha, NE 68198-5900, USA; gwang@unmc.edu; Tel.: +1-(402)-559-4176

**Keywords:** amino acid signature, amphibians, amphipathic helix, antimicrobial peptides, database, peptide design

## Abstract

Amphibians are widely distributed on different continents, except for the polar regions. They are important sources for the isolation, purification and characterization of natural compounds, including peptides with various functions. Innate immune antimicrobial peptides (AMPs) play a critical role in warding off invading pathogens, such as bacteria, fungi, parasites, and viruses. They may also have other biological functions such as endotoxin neutralization, chemotaxis, anti-inflammation, and wound healing. This article documents a bioinformatic analysis of over 1000 amphibian antimicrobial peptides registered in the Antimicrobial Peptide Database (APD) in the past 18 years. These anuran peptides were discovered in Africa, Asia, Australia, Europe, and America from 1985 to 2019. Genomic and peptidomic studies accelerated the discovery pace and underscored the necessity in establishing criteria for peptide entry into the APD. A total of 99.9% of the anuran antimicrobial peptides are less than 50 amino acids with an average length of 24 and a net charge of +2.5. Interestingly, the various amphibian peptide families (e.g., temporins, brevinins, esculentins) can be connected through multiple length-dependent relationships. With an increase in length, peptide net charge increases, while the hydrophobic content decreases. In addition, glycine, leucine, lysine, and proline all show linear correlations with peptide length. These correlations improve our understanding of amphibian peptides and may be useful for prediction and design of new linear peptides with potential applications in treating infectious diseases, cancer and diabetes.

## 1. Introduction

Amphibian species include frogs, toads, newts, and salamanders and their decline is of great concern to our society [1]. They are invaluable players in our ecosystem. There is a long history of using amphibians as human food and medicine [2]. They contain a rich source of biological molecules such alkaloids, steroids, and polypeptides. Amphibians are also important model systems for research and education [3].

Antimicrobial peptides (AMPs) are a special class of biologically active peptides [4,5,6,7,8,9,10,11,12]. They are important because of their antimicrobial effects on various pathogens, including drug-resistant bacteria, fungi, viruses, and parasites. Bombinins are one of the early amphibian peptides [12]. Peptide glycine-leucine-amide (PGLa), initially discovered in 1985 [13], was demonstrated to be antimicrobial in 1988 [14]. Magainins reported in 1987 are established as the first antimicrobial peptides from amphibians [4,15]. Since then, hundreds of amphibian peptides have been discovered. Table 1 lists the discovery timeline for the major peptide families, such as brevinins, esculentins, rugosins, temporins, and citropins [4,5,6,7,8,9,10,11,12].

The interest in antimicrobial peptides in the 1980s was driven by the desire to better appreciate immune systems by taking innate immunity into consideration [56,57]. There is also the desire to develop new types of antibiotics to meet the challenge of antibiotic resistance [9,10,11,58,59]. The advance of the AMP research is facilitated by technological development. Solid-phase synthesis [60] allowed scientists to make a sufficient amount of newly isolated peptides for detailed characterization. In the 1980s, 2D NMR method was established to determine protein structure [61,62] and timely utilized to determine the three-dimensional structure of host defense peptides in membrane-mimetic conditions [63,64,65]. It became clear that amphibian peptides adopt an amphipathic helix with distinct hydrophobic and hydrophilic faces [4,5,6,7,8,9,10,11,12]. The hydrophobic surface is flanked with basic amino acids, usually lysine. Such an interface provides a molecular basis for amphibian AMPs to target anionic bacterial membranes [6,7,8,9,10,11,12,13,14,15,56,57,58].

The bacterial killing mechanism of AMPs is complex and not well understood. It can also be concentration-dependent. At the minimal inhibitory concentration (MIC, usually at micromolar or µg/mL), a pore may form on bacterial membranes. Supporting evidence for the pore formation is the detection of ions or molecules leaked from cells after peptide treatment. Different models have been proposed to explain the pore formation. The barrel-stave model suggests that peptides assemble into a pore in bacterial membranes [66]. Shorter peptides such as gramicidin A may stack to form an ion channel [67]. As a permanent pore on bacteria is not readily observed, there was also a proposal that such pores are transient, allowing indicator dyes to enter the cell to associate with internal molecules, such as DNA or RNA. Then, the carpet model was also proposed [68]. In this model, cationic peptides bind to the anionic membrane surface. A detergent model may have a similar implication [69]. Since all these are interactions which occur in the membrane interface, the interfacial activity model was also proposed [70]. Earlier, we proposed the membrane perturbation potential for cationic AMPs based on three-dimensional (3D) structure. A peptide with more basic charges and a wider hydrophobic surface tends to be more powerful in perturbing anionic membranes [65]. Also proposed are lipid clustering [71], membrane curvature and thinning [72]. All these interfacial actions may lead to the formation of a toroidal pore with a mixture of peptide and lipids [73,74]. At even higher peptide concentrations, bacteria may be lysed and the culture becomes clear [75]. However, at a sublethal concentration where no visible damage to bacteria can be observed, the peptide, as a dangerous signal, can still upset bacteria and trigger various response mechanisms [76]. In addition, at a non-lethal concentration, peptides may also affect host cells. This results from the association of the peptide with cell receptors to trigger signal transduction events, including the release of molecules such as cytokines [59,77]. These events could be related to a variety of biological processes and therefore play a role in human health or disease.

The information on amphibian peptides is broad and scattered in different journals in the literature. To facilitate the research in the field, we entered peptide information into The Antimicrobial Peptide Database (APD). As of June 2020, 1093 amphibian peptides (1015 frog AMPs, 74 toad peptides, and 4 salamander peptides) were registered into the APD3 [78,79,80]. This article focuses on the bioinformatic analysis of frog AMPs with an attempt to identify the unifying theme behind numerous peptide families discovered and named by scientists on different continents. It will address the following questions: (1) how many AMPs were discovered per year since the publication of magainins? (2) When was the first member of each amphibian peptide family reported? (3) How will the averaged amino acid signature of amphibian peptides alter from continent to continent? (4) With so many peptide families, is there any theme that connects them? Also discussed are peptide length, charge, hydrophobicity, and post-translational modification, which are key parameters for designing potent peptides and their mimics to combat drug-resistant pathogens.

## 2. Results

### 2.1. Discovery Timeline of Frog Antimicrobial Peptides

For this study, we found 1015 frog antimicrobial peptides in the APD3, ranging from 5 to 63 amino acids (an average length of 24) [78]. However, 99.9% of these frog peptides are less than 50 amino acid residues. They have a net charge in the range of −6 to +12 with an average of +2.54. Finally, frog AMPs have a hydrophobic content that ranges from 0% to 76% (on average 51.1%). The annual total for frog peptides discovered from 1985 to 2019 is presented in Figure 1. The number of frog AMPs discovered per year steadily increased from 1987 to 2011 and then started to decline. There are two exceptional years. The first occurred in 2000 with 82 frog peptides reported. The second appeared in 2011 with 155 AMPs registered into the APD. An analysis of the database revealed that the peak in 2000 resulted mainly from the contributions of American and Australia scientists [28,30,81,82,83]. The peak in 2011 was primarily due to the contributions of Chinese scientists. As many as 73 experimentally validated AMPs from a single 2011 paper were registered into the APD [51]. It is evident that proteomic technologies greatly accelerated the discovery of amphibian peptides [84,85,86,87]. Based on the APD, we counted the total frog antimicrobial peptides contributed by scientists from different continents. A total of 503 amphibian peptides (45%) were discovered in Asia. While 151 originated from South America (14%), 179 were found in North America (18%). A total of 89 AMPs, including additional members in the magainin family, were identified in African frogs. Finally, Australians characterized 94 amphibian peptides (8%), European scientists isolated and purified 72 such peptides (6%) (Figure 2A).

### 2.2. Major Families of Frog Peptides

Figure 2B depicts the discovery timeline for the major amphibian peptide families [88] when they were first reported from different continents. Peptide source species, family counts, and reference for each family can be found in Table 1. Magainins are the first AMPs found in the African frog *Xenopus laevis* [15]. Subsequently, caerins were discovered in Australia in the 1990s [18]. With the isolation of dermaseptins in South America in 1991 [16], ranalexin was also discovered in North America in 1994 [20]. The first member of brevinin was discovered in 1992 in Asia [17]. Esculentins appeared to be the first frog AMP discovered in Europe in 1993 [19]. Additional members for some frog AMP families were subsequently discovered on different continents.

The peptide members in different families vary substantially in the APD, ranging from two to over 200. These numbers can be regarded as minimal since many predicted/isolated AMPs without antimicrobial activity data or those tested to be inactive (MIC > 100 µM) are not included in our analysis. We found 51–62 members for the ranatuerin, esculentin, and dermaseptin families. Palustrins, nigrocins, phylloseptin, and odorranains also had about 30 peptide members each (Table 1). Depending on the presence or absence of cysteine (Cys), frog AMPs can be classified into two major classes: peptides with and without Cys. A typical Cys-containing family is brevinin, which is also the largest frog peptide family in the APD with 221 peptide entries. The lengths of brevinins ranged from 17 to 37 amino acids. The majority of brevinins contain a pair of cysteines at the C-terminal region, forming a Rana box. There are also exceptions. Some brevinins contain only one cysteine, while others have five cysteines [8,51]. The odd number of cysteines in brevinin may lead to a different functional form by establishing a disulfide bond with either itself or a different molecule with an unpaired cysteine. A typical family of frog AMPs without cysteine is temporin [6]. After validation, we found 119 temporins in the APD. These peptides are relatively small with 8 to 17 amino acids. Remarkably, 72 temporins contain 13 amino acids.

Of note, cathelicidins and defensins, initially discovered in mammals [89,90], have also been identified in frogs. We found 12 cathelicidins [52] and three defensins from amphibians [55]. The discovery of these peptides enriched the reservoir of amphibian peptides.

### 2.3. Biological Activity of Amphibian Peptides

Antimicrobial activities include antibacterial, antifungal, antiviral, and anti-parasitic effects on pathogens. In addition, amphibian AMPs may have other functions such as neutralization of lipopolysaccharides (LPS), chemotactic, anti-inflammatory and wound healing [91,92,93]. The amphibian peptide counts with a variety of activities are listed in Table 2. In the following, we briefly describe these activities/functions.

*Antibacterial activity*. Magainin 2 is recognized as the first frog peptide with demonstrated antibacterial activity [4]. It is used as a model peptide for mechanistic studies. It kills bacteria rapidly. There is a consensus that this peptide acts on bacterial membranes (see Introduction). By the time this manuscript was completed, there were 982 amphibian peptides in the APD annotated to be antibacterial. These peptides usually have a broad activity spectrum and kill both Gram-positive (G+) and Gram-negative (G−) bacterial pathogens. This is one of the driving forces to develop such peptides into potential antibiotics. However, one cannot generalize this to all amphibian peptides as some members show a narrow-spectrum activity at least based on the data in the literature (below).

*AntiG-*. Gram-negative bacteria consist of both outer and inner membranes, making them more challenging to eliminate. While lipopolysaccharides (LPS) are a key component of the outer membranes, phosphatidylglycerols (PGs) and phophatidylethanolamine (PEs) are frequently found in the inner membranes of bacteria such as *E. coli* [71]. There were 84 amphibian AMPs that are mainly active against Gram-negative pathogens. Typically, *E. coli* is used as a model strain for antibacterial assays. Out of the 84 antiG- amphibian peptides, 72 are active against *E. coli*. Cationic AMPs can penetrate the outer membranes and exert their damaging effects on inner membranes [56,58].

*AntiG+*. Gram-positive bacteria do not have an outer membrane, but possess a cell wall [71]. Most AMPs are believed to target cell membranes. However, there are already AMPs that target the cell wall [78]. We found that 169 amphibian AMPs were primarily active against Gram-positive bacterial pathogens. Many temporins are such examples [5]. *Staphylococcus aureus* is often used as a model species for this assay. 122 antiG+ peptides were active against *S. aureus* and 54 were annotated to kill methicillin-resistant *S. aureus* (MRSA), usually by disrupting membranes.

*Antifungal activity*. Fungi also have a cell wall (chitin) and inner membranes (ergosterol). The cell wall components are unique and do not exist in human cells, they constitute excellent drug targets [94]. Antifungal activity could be essential to amphibians considering the recent decline. *Batrachochytrium dendrobatidis* was identified as the culprit for this decline [95]. It is found that non-declining amphibians have more effective AMPs than those declined species in the same niche [96]. A total of 474 amphibian peptides were antifungal. *Candida albicans* has been widely utilized as a model organism to test antifungal activity of AMPs. Among these peptides, 357 could eliminate *C. albicans*. In the APD, 17 amphibian peptides (e.g., magainin 2, PGLa, temporin-1P, brevinins, dermaseptin-L1, phylloseprin, and ranatuerins) are known to inhibit *B. dendrobatidis*. Amphibian peptides are likely to kill fungi by damaging membranes [8,9,10,97].

*Antiviral activity*: Viruses are challenging pathogens, as can be seen from the current COVID-19 pandemic. Viruses may, and may not, be surrounded by membranes (enveloped and non-enveloped). They can be RNA or DNA viruses. Host defense peptides such as human cathelicidin LL-37 probably plays a critical role in protecting humans from microbial infection [98]. AMPs are likely to be essential in protecting amphibians from ranaviral infection [95]. There were 45 amphibian peptides with demonstrated antiviral activity [99], mostly against human immunodeficiency virus type 1 (HIV-1) due to the interest in search of anti-HIV peptides [100,101,102]. The assay is usually conducted in viral infected cell lines by determining the viral plaque changes with and without treatment. The host cells used depend on the viral type. Caerin 1.1, caerin 1.9, and maculentin 1.1. are demonstrated to inhibit HIV infection rapidly. They appear to inhibit the initial stage of viral fusion with host cells [100]. Of note is that frog urumin is shown to specifically inhibit influenza A virus H1N1 via binding to hemagglutinin [103]. Temporin B is found to inhibit herpes simplex virus 1 (HSV-1) by disrupting viral membranes. It could influence other stages of viral infection ranging from attachment to replication [104]. The potential multiple hits could reduce the resistance development of pathogens.

*Antiparasitic activity*: Protozoa are in the category of neglected tropical diseases (NTDs). Malaria is one of such diseases. Due to the complex life cycle and rapid mutation of parasites, it remains challenging to eradicate such pathogens. Amphibian AMPs also have an effect on parasites in infected cells [105]. In the APD, 53 amphibian peptides were demonstrated to have activity against parasites. Dermaseptin 01 is able to lyse most of the protozoan cells after incubation for 2 h [106]. Dermaseptins also showed anti-malarial potency with 50% growth inhibitory concentrations (IC_50_) in the micromolar range [107]. There are different strains responsible for human malaria, but *Plasmodium falciparum* is most virulent [108]. In the APD, 23 peptides, including amphibian magainin 2 and dermaseptin-S4, are known to inhibit *P. falciparum*. A natural concern is the eukaryotic nature of parasites, which may limit cell selectivity. However, it is found that membranes of infected human red blood cells more resemble parasites and contain increased amounts of anionic phosphatidylinositol (PI) and phosphatidic acid (PA)*,* which may endow the desired selectivity [108].

*Anticancer activity*. There is a high interest in the anticancer effect of AMPs [98,109,110]. In the APD, 93 amphibian peptides were demonstrated to have an effect on cancer cells. The classic idea is that cancer cells are transformed and have anionic phosphatidylserine (PS) exposed. Such a bacteria-like membrane property may provide a molecular basis for cell selectivity of cationic peptides. However, we found limited cell selectivity for frog temporins with high hydrophobicity [111]. There are also other anticancer mechanisms. Dermaseptin-PS1 inhibits human glioblastoma U-251 MG cells (from cells frozen in Uppsala in 1969) via inducing apoptosis at a low micromolar concentration and disrupts cancer cell membranes at 10-fold the concentration [112]. Moreover, brevinin-1GHd was shown to inhibit proliferation of human cancer lines [113]. As a means to improve cell selectivity of anticancer peptides, some compounds may be utilized to potentiate peptide potency [114].

*Anti-diabetic activity*. Type 2 diabetes (T2D) is a condition where insufficient insulin is released to keep the glucose balance in a human body. It is hypothesized that stimulating the production of insulin may provide an avenue of treatment. Amphibian peptides can induce the release of insulin from β-cells. These peptides may be useful to treat T2D. In the current database, 15 amphibian AMPs were annotated to have this property [115,116]. Further studies are ongoing to improve peptide selectivity and stability of amphibian peptides and their analogs [116]. This will pave the way to test in vivo efficacy of the promising candidates in animal models.

*Insecticidal activities*. Pesticides play an important role in suppressing pests harmful to crops. They also caused environmental pollution and are perhaps part of the reasons for the decline of amphibians. In this regard, alternatives are sought to minimize the impact of pesticides on our environment. Naturally occurring AMPs may be sprayed to kill pests. Alternatively, frog AMPs, when expressed in plants, also showed insecticidal effects [117].

*Spermicidal activities.* AMPs may also be developed into novel contraceptives. A few dermaseptins were found to have spermicidal activity [118,119]. This is because anionic sulfogalactosylglycerolipid (SGG) and sulfogalactosylceramide (SGC) in the head region of sperms make the surface negative and can become a preferred target of AMPs [120]. Ongoing research is attempting to improve peptide selectivity and stability to avoid the cleavage by host proteases.

*Endotoxin neutralization, immune modulation and wound healing*. Some frog peptides such as temporins [121] are known to associate with LPS (endotoxin), which can regulate the release of cellular cytokines and lead to an anti-inflammatory effect [122]. Most of these effects can promote wound healing [91,92,93].

*Antioxidant and protease inhibitory activities*. In the APD, 19 amphibian peptides were found to have antioxidant effects [51]. Notably, some frog AMPs can inhibit proteases [84]. The protease inhibitory properties of amphibian AMPs is remarkably interesting and deserves further study. Is this property necessary for antimicrobial activity? If not, which functional role requires the stability of these peptides in innate immune systems?

*Synergistic effects between amphibian AMPs*. We have assumed that one peptide performs the above tasks. Some amphibian AMPs such as uperin 3.5 are capable of forming amyloid fibrils in solution at pH 7, which are toxic to neuronal cells. Since this peptide has the tendency to form different oligomers, these aggregates may play a role in bacterial membrane damage [123]. Moreover, a single peptide chain may be covalently linked via a disulfide bond to achieve other advantages. The dimerization of distinctin to a helix bundle confers stability to the peptide as the monomer is equally active [34]. Furthermore, two different peptides (e.g., PGLa and magainin 2 from the same African clawed frog *Xenopus laevis*) can work together to produce a synergistic antimicrobial effect. The most recent results suggest the formation of heterodomains and enhanced peptide affinity toward bacterial phosphatidylethanolamine [124]. Such an assembly and synergy of amphibian peptides further widens our view on innate immune peptides and suggests the benefit of combined therapy.

### 2.4. Toxicity of Amphibian Peptides

Identification of amphibian peptides with the antimicrobial activities or other functions above is an important step. To develop any peptide for therapeutic use, it is essential to know its toxicity to mammalian cells. Ideally, the peptide should be not toxic. Hemolysis of red blood cells (RBC) is usually the first test since it is convenient to conduct. Different blood cells are used in the literature, adding difficulty to compare cytotoxicity of these peptides. We recently compared several types of blood cells and the results are similar to those obtained from human RBCs [125]. A more thorough toxicity analysis requires the use of different host tissues. In our recent study, we compared the toxicity of the same peptide on kidney, lung, spleen, and liver cells [126]. More importantly, the Food Drug Administration of the United States (FDA) requires toxicity evaluation of the lead compound in at least two different animal models. Our recent toxicity evaluation of designer antimicrobial peptides in both mice and rats provides an example for this [126].

We have annotated hemolysis of AMPs in the first version of the APD [80]. It may be reasonable to set 100 µM as a threshold for toxicity, but there is no consensus. Perhaps, a definition of the required cell selectivity is more useful considering the differences in MIC values of AMPs. Cell selectivity is usually defined as the ratio of 50% hemolytic concentration (HC_50_) and MIC. At present, 163 amphibian AMPs were annotated to be hemolytic. Because peptide hydrophobicity is known to be important for hemolysis, we calculated the averaged hydrophobic content of these peptides as an approximation. Indeed, the total hydrophobic content (55.6%) of the 163 hemolytic peptides is higher than that (51.1%) summed for all the 982 antibacterial peptides, while both averaged net charge and lysine% are similar (Table 3). Such a picture has not changed since we observed the higher hydrophobic content for hemolytic peptides in our original database than any antimicrobial group [80]. This database observation is in line with our discovery that reducing hydrophobicity is a general avenue for improving cell selectivity of membrane-targeting cationic peptides [98,127].

### 2.5. Peptide Sequence Signatures Modulate Activity Spectrum of Frog Host Defense Peptides

We first defined frequently occurring (abundant) amino acids in the signature of AMPs (i.e., a profile of 20 amino acids for one or more peptides, see Figure 3) [79]. Amino acids leucine (L), alanine (A), glycine (G), and lysine (K) are frequently occurring in amphibian AMPs [128]. Our classification of the amphibian peptides from different continents revealed subtle differences in the proportions of these amino acid residues [129]. For instance, alanine is more abundant in frog peptides from South America, while leucine is slightly higher in peptides discovered in Europe. To validate our observation, we also analyzed 221 brevinins discovered in Asia, Europe and North America (Figure 3). The European brevinins also had a higher level of leucine (16%) than the two groups from Asia and North America. In contrast, alanine was higher in frog AMPs from North America (14.4%) than those from Europe (9.4%). It is possible that the amino acid signatures for some frog AMPs deviate from those of brevinin. On average, temporins (8–17 residues) are shorter than brevinins (17–37 residues) [5]. They have been identified in Asian (66 peptides), African (5 peptides), European (19 peptides), and North American frogs (30 peptides in Figure 4). Surprisingly, temporins were especially abundant in leucine with the highest (38%) from European peptides and the lowest (25%) from African amphibian peptides. In fact, all 119 temporins in the APD contain at least one leucine. Both alanine and lysine had reduced levels in this special peptide family (below 10%). The variation in the levels of these four residues (L, A, G, and K) in different peptide families modulates peptide activity. While 45% of temporins were active only against Gram-positive bacteria, only 5% brevinins were annotated to have the same activity spectrum. There are also common features in the sequences of temporins and brevinins. A hydrophobic cluster or motif, usually containing a proline at position 3, exists at the N-terminus of 67 temporins and 128 brevinins, while charged residues (usually lysine) are located in the center or both the center and the C-terminus. The hydrophobic amino acids (X) involved in this cluster XXPXX are usually leucine (L), isoleucine (I), valine (V), phenylalanine (F), and occasionally alanine (A). Frequently, the second X is a leucine. Based on our recent structural analysis [111], the proline in this *N*-terminal motif introduces a bend so that all the four hydrophobic amino acids come together to extend the hydrophobic surface of the three-turn helix.

### 2.6. Length-Dependent Correlations in Frog Antimicrobial Peptides

The change in the amino acid signatures for AMPs with different lengths (brevinin vs. temporins) (Figure 3 and Figure 4) prompted us to investigate the factors that connect different frog peptide families. We reasoned that frog AMPs would be ideal for this study since (1) there is a large number (over 1000) in the APD and (2) they are relatively homogenous in sequence, usually forming an amphipathic helix [4,5,6,7,8,9,10,11,12]. To get insight into the length-dependent design of frog AMPs, we separated these peptides into numerous length groups at a step size of five (e.g., 6–10, 11–15, 16–20, 21–25, 26–30, 31–35, 36–40, 41–45, and 46–50). The last group is 46–50 because 99.9% of these frog AMPs in the APD have a peptide length less than 50 amino acids. We excluded the 41–45 length group from this analysis because it is a small group with only four peptides and contains the only two frog defensins, which are drastically different from linear helical peptides. The averaged net charge and hydrophobic content for each group of frog AMPs were then calculated by using the database statistical analysis after each peptide search [80]. In the APD, the hydrophobic content (Pho) is the sum of the percentages for leucine, isoleucine, valine, methionine, alanine, cysteine, phenylalanine, and tryptophan. Figure 5 shows the charge-length and Pho-length plots. With an increase in peptide length, we observed a proportional increase in peptide net charge (Figure 5A). On contrary, the hydrophobic content decreased with an increase in peptide length (Figure 5B). In other words, longer frog peptides tend to have more charged amino acids at the expense of hydrophobic amino acids. To illustrate, the long frog peptides in the APD are mostly esculentins with 46–49 amino acids, an averaged net charge of +5.8 and a hydrophobic content of 41.5%. On the opposite side, the short peptides with 11–15 amino acids are mostly temporins and aureins with an average peptide length of 13.2, hydrophobic content of 59.8% and net charge of +1.0.

We then asked which amino acids are proportionally changed with peptide length, leading to the linear correlations in Figure 5. For this purpose, we also calculated the average amino acid contents for each amino acid in the different length groups at the same step size of 5. We scanned through the plots of the residue content versus peptide lengths for all the 20 amino acids. The top four amino acids that displayed a linear relationship are plotted in Figure 6. They are (A) leucine; (B) glycine; (C) proline; and (D) lysine, three of which are the frequently occurring amino acids of frog AMPs mentioned above. This is interesting considering leucine, glycine, and lysine are sufficient to design active peptides [39]. While both glycine and lysine are proportional to peptide length, leucine and proline are inversely proportional to peptide length. The correlation coefficients (R^2^) for these four plots are in the range of 0.73–0.90 (Figure 6), indicating a linear relationship. Moreover, phenylalanine shares a similar inverse relationship with leucine, although the correlation coefficient is 0.62 (not shown). The next amino acid with a correlation coefficient greater than 0.5 is cysteine. The rest of the amino acids are less correlated with R^2^ ranging from 0.009 to 0.45. It is likely that the decrease in proline in longer peptides is to retain the helical structure required for peptide function. We propose that these linear relationships uncovered herein in Figure 5 and Figure 6 might be utilized to design new linear peptides at different lengths. To illustrate this, we analyzed 62 dermaseptins (DRS), one family of amphibian peptides discovered mostly in South America [5]. These peptides were separated into three length groups (21–25, 26–30, 31–35 amino acids) and the features of each group are listed in Table 4. It is remarkable that there is a proportional increase of alanine, lysine, and net charge with peptide length, while leucine and glycine are inversely proportioned to peptide length in dermaseptins. These relationships are essentially the same as we found above based on all frog peptides. The only exception is glycine. This may result from the abundance of a small alanine, which can be regarded as an analog of glycine (Table 3). Indeed, many dermaseptins start with either the GLW or ALW sequence motif. W3 is a highly conserved aromatic residue in dermaseptins, facilitating peptide quantification by UV [5]. Our analysis here in Table 4 uncovers the design parameters of dermaseptins at various length groups, yielding novel insight into these amphibian peptides.

## 3. Discussion

Amphibians produce numerous active peptides for a variety of biological functions such as antioxidant, hormone, growth factor, antibiotics, immune modulation, and wound healing [8,51] [visit also Database of Anuran Defense Peptides (DADP); http://split4.pmfst.hr/dadp/]. This study focuses on amphibian peptides with demonstrated antimicrobial activity collected in the APD database. However, amphibian AMPs can have other functions such as anticancer, anti-diabetes, and spermicidal functions (Table 2). The expression of such peptides is not understood, but can depend on pathogen types, amphibian species [4,51], sex [130], life stages [131], season [130,132], and geographic regions [133]. Cultivation of frogs under sterilized conditions reduces AMP expression and can be restored after exposure to the natural environment [133]. It is clear that amphibian AMPs discovered from different frog species are rather different. Occasionally, the same sequence was “found in multiple species”. A search of the APD using the quoted phrase returned 56 shared amphibian AMPs. This is relatively small compared to the total of 1093 amphibian AMPs examined during our analysis. Male and female frogs may not express the same set of AMPs. For example, caerin 1.10 is expressed in the male Australia magnificent tree frog, *Litoria splendida*, but not in female [130]. The level of such peptides also varies with season [130,132]. Cathelicidin-Bg peaks in August and September [132]. Modern systems biology, genomics, proteomics, lipidomics, metabolomics and microbiota aim at integrating such information for a more complete understanding of peptide functional dynamics in a defined ecological niche. In particular, peptidomic studies led to the discovery of up to 100 peptides in a single frog, which presumably conduct a variety of functions beyond antimicrobial activity. This study shines light on frog AMPs from the angle of bioinformatics. Our study was made possible due to our continued registration of the peptide data into the APD database for over a decade. The reliability of the data has been increasing with time, owing to the establishment of peptide registration criteria and its continued updating. Our analysis reveals that four amino acids are abundant in amphibian AMPs: leucine, alanine, glycine, and lysine. These amino acids are useful in designing new AMPs [79]. The variations of such amino acids modulate activity of different peptide families and may offer a living advantage on a continent [129]. In the following, we discuss peptide length, hydrophobicity, charge, and post-translational modification, which are the key parameters for designing AMPs.

### 3.1. Peptide Length

There appears to be a minimal requirement for peptide length in order for it to be antimicrobial. During isolation of novel AMPs, some truncated peptides were found to be inactive [8,134,135]. Likewise, further truncation of the minimal antibacterial peptide KR-12 of human cathelicidin LL-37 led to inactive peptides such as RI-10 [136]. On a different track, we found a non-toxic bacterial membrane-targeting sequence (Figure 7A) [137] could be converted to an antibacterial peptide when D13 was changed to F13 [65] based on sequence homology to the antibacterial and anticancer peptide aurein 1.2 (Figure 7B), isolated from an Australia frog [28]. However, some amphibian AMPs, such as esculentins, are much longer and can be up to 50 residues. Little is known why and how frog AMPs are constructed at varying lengths. In terms of the simplicity principle, shorter peptides would be advantageous to reduce synthesis cost. Then why bother with long peptides? One possibility is that the short and long peptides confer different antimicrobial capabilities. In particular, longer peptides may be able to form two structural domains that work synergistically against the pathogen membranes. Maximin 4 has a helix–break–helix structure in either micelles or organic methanol [138]. A different model was found in human cathelicidin LL-37 (Figure 7C), where a long helix is separated into two hydrophobic domains for synergistic binding to bacterial LPS [136,139]. Another possibility is that long sequences can encode additional sequence elements for specific molecular recognition that triggers signal pathways for the functional regulation and coordination required for the survival and health of frogs. Future studies may elucidate the exact functions of these peptides in anurans, including a possible role as poisons against predators [140].

### 3.2. The Ratio of Hydrophobic and Basic Amino Acids Determines Peptide Activity spEctrum

Once the peptide length is decided, both hydrophobic and cationic amino acids are considered to generate the popular amphipathic sequences of amphibian AMPs [4,5,6,7,8,9,10,11,12]. Our recent analysis of all the AMPs in the APD uncovered a linear relationship of the peptide hydrophobic content (Pho) with the percentage of arginine, but not lysine [126]. However, there is a clear linear relationship between lysine and Pho when Pho is below 50% in the plot. We found here different requirements for hydrophobic contents depending on the bacterial Gram type. The hydrophobic content is higher for AMPs against Gram-positive bacteria than Gram-negative bacteria (Table 3). Leucine is a frequently occurring hydrophobic amino acid that can be used to represent all hydrophobic amino acids. In addition, phenylalanine (Phe) appears to be an important aromatic residue to anchor such peptides onto membranes. In several cases (Figure 7), we have demonstrated direct magnetic dipole–dipole interactions between aromatic rings of Phe and anionic phophatidylglycerol (PG) [65,136,141]. This may explain why Phe increases with a decrease in peptide length in helical peptides [129]. A typical example is temporin-SHf (sequence: FFFLSRIF) [142], an eight residue peptide with 50% Phe. The four Phe residues in this natural frog peptide are reminiscent of human cathelicidin LL-37. The four Phe aromatic rings (F5, F6, F17, and F27) of LL-37 (Figure 7C), as well as arginine, all interact with anionic lipid PG [136]. Such features are remarkably conserved in temporin-SHf, making it a minimal LL-37 like peptide. In contrast, the averaged net charge is slightly higher for peptides active against only Gram-negative bacteria than those against Gram-positive bacteria. This basic amino acid is usually lysine in frog peptides. That explains why lysine can be utilized to represent basic amino acids in amphibian AMPs in general. This database observation is in line with our results obtained from the structure–activity relationship study of a broad-spectrum helical peptide GF-17, the major antibacterial peptide of LL-37 [127]. A decrease in peptide hydrophobicity by disrupting the helical backbone or by sequence truncation eliminated its activity against Gram-positive *S. aureus* but not Gram-negative *E. coli*. However, conversion of three arginines to alanines in 17BIPHE2 (a stable, selective and potent peptide designed based on GF-17) made the peptide active against *S. aureus* but not *E. coli* [128,143]. Hence, the ratio between hydrophobic and charged amino acids is a determinant of the peptide activity spectrum (Table 3). These results may open the door to engineering peptides with a desired activity spectrum to selectively eliminate the unwanted pathogens with no harm to commensal bacteria.

### 3.3. Hydrophobic and Basic Amino Acids Change with Peptide Length

This study shines light on the deployment of charged and hydrophobic amino acids in amphibian peptides with varying peptide lengths. With an increase in the peptide length of anuran peptides, there is a decrease in peptide hydrophobic content and an increase in averaged net charge (Figure 5). Moreover, the decrease in peptide hydrophobic content is related to a decrease in leucine. In contrast, the increase in net charge with an increase in peptide length is mainly contributed by lysine (Figure 6). The increase in glycine with peptide length may be important for the peptide flexibility required for peptide activity. The decrease in proline in long peptides is likely to avoid the loss of the amphipathic helical structure critical for function. Proline can confer special structural and biological function. In buforin, the proline appears to be critical for the peptide to enter the cell [144]. In a hybrid cecropin–magainin peptide, P18, a central proline is critical for antifungal activity rather than a more helical structure [145]. The linear relationships discovered herein (Figure 5 and Figure 6) provide one mode to unify a variety of amphibian peptides (Figure 2B). These relationships might be useful to design new peptides at different lengths. It is relevant to point out the limitations of these relationships derived from linear anuran peptides, including those stabilized by a Rana box. Anionic amphibian peptides (16 in the APD) do not obey these equations. Moreover, these relationships are not applicable to non-linear antimicrobial peptides, such as multiple disulfide-bonded defensins and circular cyclotides [89,146,147]. Those peptides are more complex and restricted in sequence length and amino acid composition (summarized in ref. [78]).

### 3.4. Post-Translational Modification Improves Helix Stability and Activity

As a fourth parameter to understand amphibian AMPs, post-translational modification can also modulate peptide structure and activity [148]. Although 24 types of chemical modifications have been annotated in the APD3 [78], such modifications are very limited in amphibian peptides [51,84,85,86,87]. Here we only describe C-terminal amidation and Rana box. C-terminal amidation results from chemical modification of the last glycine in the peptide. In the APD database, 35% of amphibian AMPs are C-terminally amidated to increase the peptide net charge by +1 (for accuracy, this modification is considered in the APD when calculating the net charge of peptides). C-terminal amidation can be essential for the antimicrobial activity of amphibian peptides [133,149]. A plausible reason is stabilization of the helical structure to allow for the formation of one additional hydrogen bond. In addition, 470 amphibian peptides in the APD contain a Rana box, usually at the C-terminus where 4–6 residues are bracketed between a pair of cysteines that presumably form a disulfide bond [5,6,7,8,9,10]. This disulfide bond can stabilize the helix and may be a prototype for stapled helices [150]. Disruption of the Rana box can reduce peptide activity. Note that both C-terminal amidation and a Rana box can co-exist in the same frog peptide (seven such peptides in the current APD). It is likely that the Rana box is broken in reduced physiological conditions. Then, the free cysteines can enhance anti-oxidant ability of the peptide [149] or perform other functions yet to be elucidated.

### 3.5. Peptide Design

For decades, scientists have been attempting to understand nature’s design principles for amphibian AMPs [4,5,6,7,8,9,10,11,12] so that we can design useful peptide therapeutics. Focus has been placed on designing a new generation of antibiotics to overcome the microbial resistance problem. The rapid killing, broad antimicrobial spectrum and low chance of resistance development are regarded as advantages over conventional antibiotics. However, narrow-spectrum peptides are also of outstanding interest nowadays in the era of microbiota. To design an effective peptide, one needs to determine its length, charge, hydrophobic content, and chemical modification. It is to our advantage that the helical backbone can be decorated with different amino acids to confer the desired antimicrobial activity or mechanism of action by varying the type and ratio of charged/hydrophobic amino acids [151]. In addition, the classic helical structure may be distorted locally or globally to improve protease stability and to reduce peptide toxicity without a substantial loss of antibacterial activity [127,152]. The peptides made entirely using D-amino acids are known to be more resistant to proteases [56]. Magainins have been extensively explored as potential membrane-targeting antibiotics [153]. Buforins may be a useful template to design peptides to attack intracellular pathogens [144]. The N-terminal region (residues 1–18) of esculentin-1A has also been investigated for various applications [152,154,155]. Using a computing approach, Juretić and colleagues designed a selective peptide [156]. Quantitative structure–activity relationship (QSAR) studies are also applied to the design of antimicrobial and antibiofilm peptides [157,158]. We developed database-guided methods [159]. Although our database screening identified potent peptides against MRSA or HIV-1 [102,160], it is more inspiring to develop the database filtering technology [161]. This technology illustrates nicely how to derive the above discussed peptide parameters step-by-step by following the most probable principle. Since the target pathogen was MRSA, we arrived at a new peptide template with a high hydrophobic content and low cationicity. Remarkably, our database designed DFTamP1 peptide kills MRSA both in vitro and in vivo. However, insertion of additional lysines to the optimized DFTamP1 peptide is detrimental to in vivo efficacy [162]. Our results reveal the importance of balanced charged and hydrophobic residues for systemic efficacy, implying potential systemic uses of certain amphibian peptides with special properties.

### 3.6. Peptide Mimicries

Along the classic path to antibiotic development, there is the desire to make peptides stable so that they can work longer in the presence of host/pathogen proteases. Deployment of multiple Rana box like stapled motifs along the helix is a state of the art design to confer stability to helical peptides [150]. Distortion of the peptide backbone can also confer stability [68,127]. As an alternative strategy, peptide mimics, such as oligomers of acylated lysines (OAKs) and arylamide oligomers, are made to improve peptide stability (reviewed in refs [163,164,165]). They can be normal α-peptide analogs, β-peptides or their combinations [166], or non-peptide polymers [163,167]. It is also possible to make small molecules to mimic cationic AMPs. DFTamP1 mimics are such examples [168]. The successful synthesis of a variety of mimics underscores that the classic regular amphipathic structure is not a must for antimicrobial activity. Irrespective of peptide scaffold or size, however, both charged and hydrophobic moieties are deployed in peptide mimics. Systematic studies of these mimics enabled the identification of lead candidates for in vivo tests. In addition, one can consider numerous application strategies to enhance the chance of success. These include (1) induction of the AMP expression in the host, (2) the use of probiotic microbes to deliver the peptides, (3) the design of the prodrug, (4) the construction of peptide conjugates to improve cell targeting, and (5) combination with conventional antibiotics to improve efficacy and to reduce toxicity [169].

## 4. Methods: Database, Data Collection Criteria and Analysis

### 4.1. Database Versions and Update

This study was conducted based on the APD (website: http://aps.unmc.edu/AP; accessed June 2020). Our database project was initiated in 2002 as the thesis work of a graduate student in our laboratory. The first version of our database with 525 entries was published in 2004 [78]. The second version reported 1228 peptide entries in 2009 [79]. The number of AMPs increased to 2619 in the APD3 [80]. The APD database was continuously updated [129]. As of 29 June 2020, there were 3201 AMPs in this database, mainly from natural sources.

### 4.2. Activity and Structural Informational Annotations

In the literature, antimicrobial activity can be obtained by different methods such as broth dilution and the agar diffusion method [170]. The microdilution method is most widely utilized due to convenience. The results are expressed as MIC in either micromolarity (µM) or µg/mL when bacteria do not grow in the 96-well microplates. The advantage of micromolarity over µg/mL is that it allows us to compare the activity of peptides at different sizes. For those determined by the diffusion method, they will be accepted as long as they are active below the defined concentration. Because these AMPs are tested by different laboratories under different media conditions, it poses a challenge to classify them into strong, medium and weak on a universal activity scale. The APD is attempting to assign “strong” to AMPs if the MIC values are below 10 µM against all testing bacterial strains and “weak” if the MIC values are above 50 µM. Finally, all the activity annotations were based on the literature data. Therefore, the activity spectrum for each peptide entry could change with time when more data become available.

The APD also annotated structural information for amphibian peptides during 2002-2003 [80]. Circular dichroism (CD) is frequently utilized to provide secondary structural information. As a helix has a characteristic feature, we included CD information in the APD3 [78] as evidence for helix. To date, all the 3D structures of amphibian peptides in the APD were determined by two-dimensional (2D) nuclear magnetic resonance (NMR) spectroscopy in membrane-mimetic environments such as organic solvents and micelles. The structure can be directly viewed via the Protein Data Bank (PDB at https://www.rcsb.org/) link in the APD. When multiple structures are available, the link usually points at the better-defined structure to reduce redundancy.

### 4.3. Peptide Classification

The biological sources of AMPs in the APD3 are systematically separated into six kingdoms: bacteria, archaea, protists, fungi, plants, and animals [78]. The majority of the peptides originate from the animal kingdom (74%) and 1093 amphibian AMPs account for almost half of animal peptides in this database. In 2010, peptide 3D structures were unified into four classes: α, β, αβ, and non- αβ based on the presence or absence of an α-helix and a β-sheet in the peptide structure [147]. In the case of amphibians, it is interesting to note that all the currently known 133 structures are α-helical (determined by solely by NMR or implied by CD). A unified peptide classification scheme that does not depend on peptide source, 3D structure, and activity was also introduced into the APD3 in 2015 [147].

### 4.4. Data Registration Criteria and Analysis

Many anuran peptides were isolated from skin secretions after electrical stimulation. In the first two versions of our database, we attempted to register all the antimicrobial peptides into the APD, including peptides without activity data. With the application of genomic and peptidomic approaches, 728 peptides were reported in one paper from nine frog species (81 each species) [51]. It became a question whether all the 728 peptides should be registered into the APD. The antimicrobial testing results in that paper gave us the answer. Nearly all the isolated peptides show a good activity, whereas chemically synthesized predicted peptides are largely not antimicrobial [51]. To further verify this, we synthesized selected amphibian peptides, which were previously isolated from frogs without activity data due to a limited quantity, and tested their antibacterial, antifungal, and anticancer activities. These isolated peptides are indeed antimicrobial [111]. In addition, colleagues have noticed that not all amphibian peptides are antimicrobial [5,6,7,8,9,10,11,12]. A peptide might have been isolated earlier, but its antimicrobial status will not be conferred until the antimicrobial activity is determined [4]. As a consequence, the APD currently registers only peptides with demonstrated antimicrobial activities (MIC < 100 µM or 100 µg/mL) [78]. Consequently, some old amphibian peptide entries were replaced during this study due to MIC values greater than 100 µM. Our rigor in data registration led to a reliable data set. The majority of the peptides are linear, helical [64,151], and relatively homogeneous, making the results of this analysis useful for peptide design and prediction. The AMP analysis was conducted in the APD and linear regression was performed using Excel to search for correlated peptide parameters. Figures were made using Excel/Powerpoint and Figure 7 using MOLMOL [171].

## 5. Conclusions

Amphibian peptides are diverse. As of June 2020, 1093 amphibian antimicrobial peptides were registered in The Antimicrobial Peptide Database. These peptides are predominantly linear and usually fold into a classic amphipathic helix in membranes. The deployment of different proportions of hydrophobic and cationic amino acids modulates peptide activity. Thus, it is likely that not all amphibian peptides display a broad-spectrum antimicrobial activity (Table 2). Peptides with a high hydrophobicity and low cationicity tend to be potent primarily against Gram-positive pathogens such as MRSA. Narrow-spectrum peptides can be important for future personalized antibiotics that selectively eliminate pathogens of concern. Our bioinformatic analysis of the 1015 frog peptides uncovers multiple length-dependent correlations. Short frog peptides tend to be more hydrophobic and less charged. With an increase in peptide length, charged lysine and glycine increase, whereas hydrophobic residues such as leucine and phenylalanine decrease in frog peptides. Proline, a helix breaker, is less common in longer peptides discovered in frogs. These relationships may be utilized to design new types of peptides with different sizes. They may also guide the synthesis of peptide mimics such as polymers and small molecule mimics. Such relationships may also be included in computer programs to predict new AMPs or to design new antibiotics to combat drug-resistant pathogens. Since amphibian peptides also possess other functions, the designed peptides may find use as immune regulators, anticancer and anti-diabetic drugs.

## Figures and Tables

**Figure 1 antibiotics-09-00491-f001:**
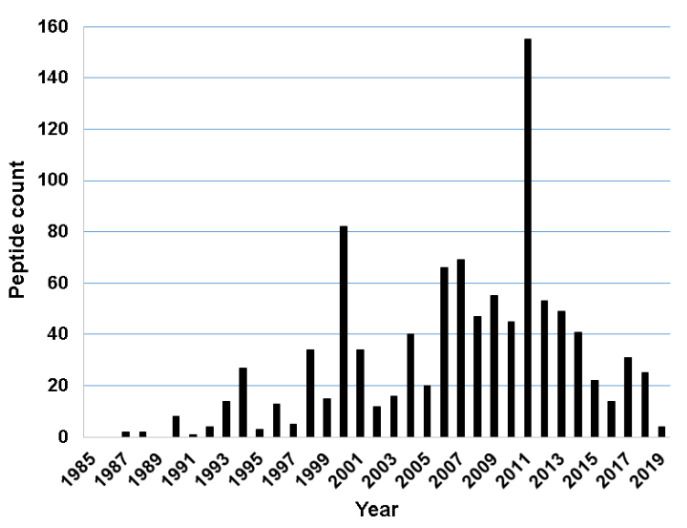
Frog antimicrobial peptides discovered annually from 1985 to 2019. Data in all the figures presented in this study were obtained from the Antimicrobial Peptide Database (http://aps.unmc.edu/AP) in June 2020.

**Figure 2 antibiotics-09-00491-f002:**
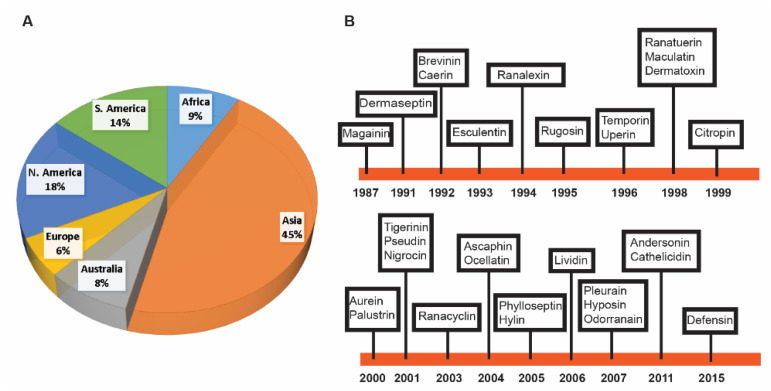
(**A**) Percentages of amphibian antimicrobial peptides discovered on different continents based on the APD (http://aps.unmc.edu/AP) as of June 2020; (**B**) Discovery timeline for major amphibian antimicrobial peptide families.

**Figure 3 antibiotics-09-00491-f003:**
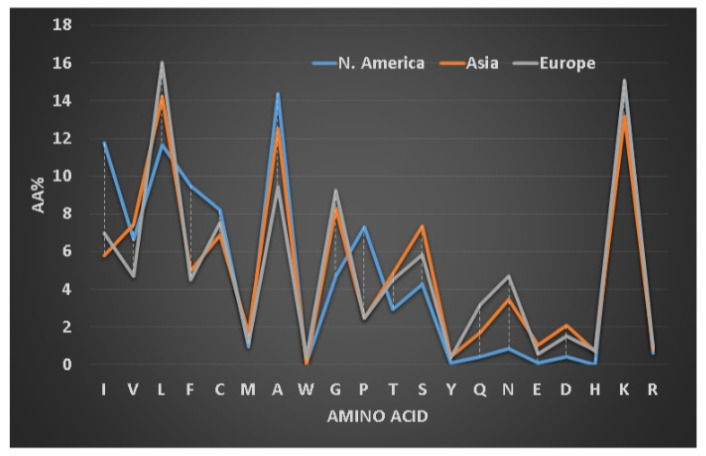
Variation in the amino acid signatures of brevinin peptides discovered in three regions: 152 from Asia, 20 from Europe and 48 from North America. The dotted vertical line indicates the maximal difference between two groups. Data from the APD as of June 2020 [78].

**Figure 4 antibiotics-09-00491-f004:**
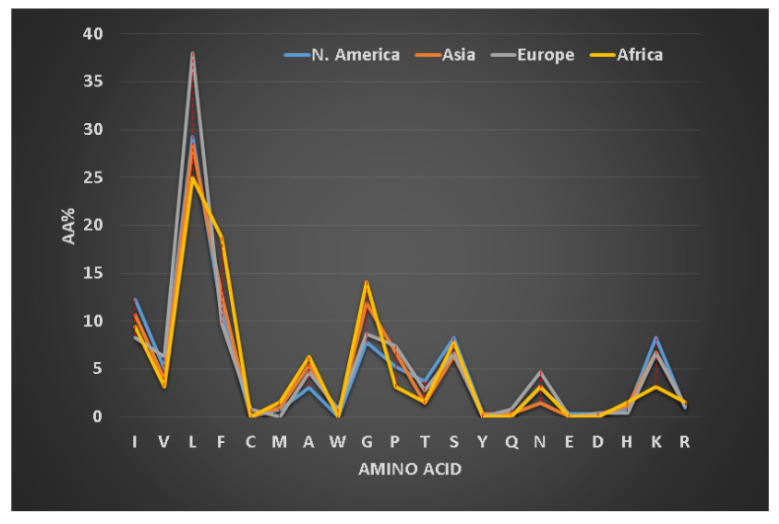
Variation in the amino acid signatures of temporins discovered in four regions: 5 from Africa, 66 from Asia, 20 from Europe and 30 from North America. The dotted vertical line indicates the maximal difference between two groups. Data from the APD as of June 2020 [78].

**Figure 5 antibiotics-09-00491-f005:**
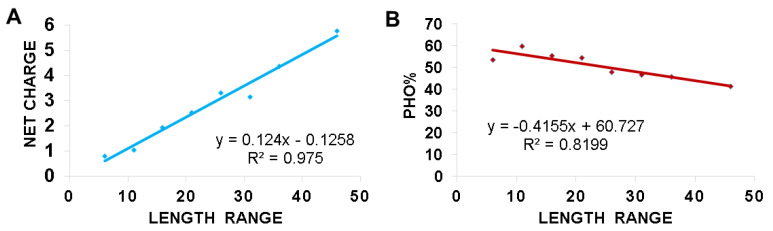
Relationships between peptide length and averaged net charge (**A**); and total hydrophobic content (**B**) based on the data in the APD. The 1015 frog antimicrobial peptides in the APD were split into various groups at a length step size of 5: 6–10, 11–15, 16–20, 21–25, 26–30, 31–35, 36–40, 41–45, and 46–50 amino acids. The number of peptides in each group varies from 10 to 325. The length group 41–45 was not included in this plot due to a small number of only four members and the existence of the only two frog defensins. Data from the APD as of June 2020 [78].

**Figure 6 antibiotics-09-00491-f006:**
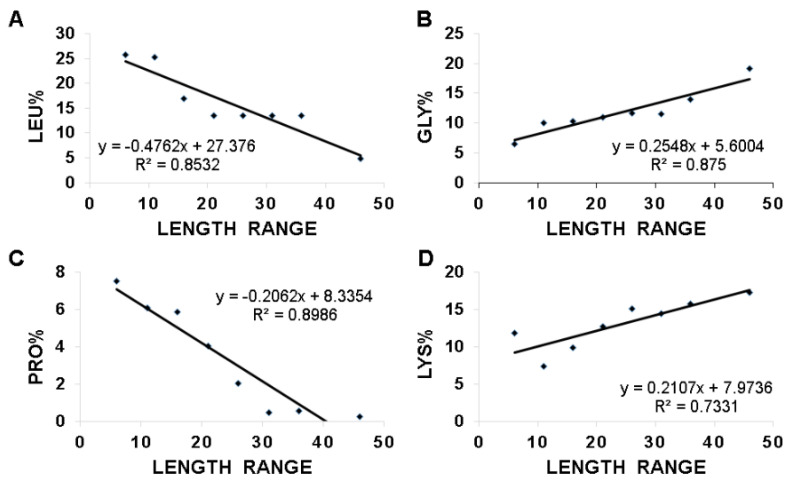
Relationships between peptide length and the average amino acid content of leucine (**A**); glycine (**B**); proline (**C**); and lysine (**D**) based on the data in the APD as of 29 June 2020. The 1015 frog antimicrobial peptides were split into various groups at a length step size of 5: 6–10, 11–15, 16–20, 21–25, 26-30, 31–35, 36–40, 41–45, and 46–50 amino acids. The number of peptides in each group varies from 10 to 325. The length group 41–45 was not included in this plot due to a small number of only four members and the existence of the only two frog defensins. Data from the APD as of June 2020 [78].

**Figure 7 antibiotics-09-00491-f007:**
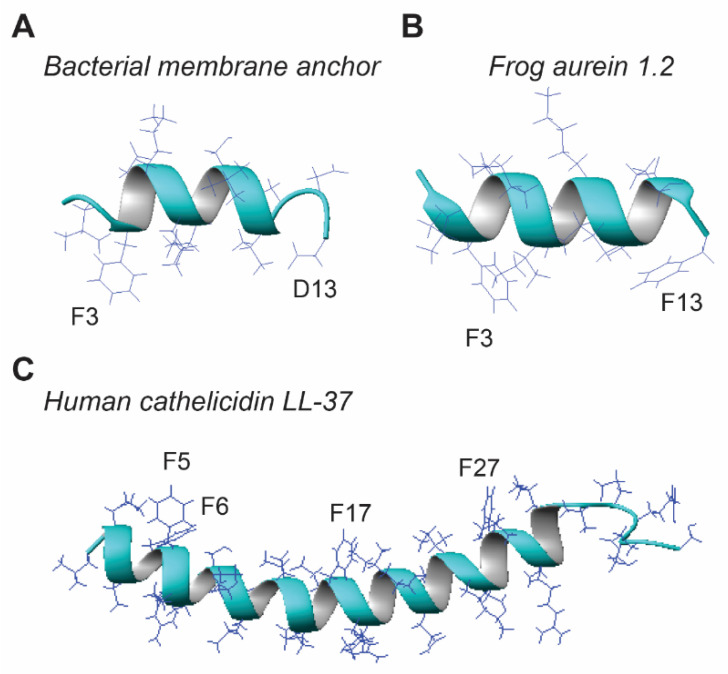
Three-dimensional structures of (**A**) *E. coli* membrane anchor; (**B**) amphibian aurein 1.2; and (**C**) human cathelicidin LL-37 determined by multidimensional NMR spectroscopy [65,136,141]. The importance of the aromatic rings for membrane targeting is indicated by an intermolecular nuclear Overhauser enhancement (NOE) between the aromatic protons and bacterial anionic phosphatidylglycerols (PGs).

**Table 1 antibiotics-09-00491-t001:** Discovery timeline of the major families of frog antimicrobial peptides ^1^.

Year	Peptide	Source Species	Continent	Count	Ref ^2^
1987	Magainin	*Xenopus laevis*	Africa	8	[15]
1988	XPF	*Xenopus laevis*	Africa	13	[14]
1991	Dermaseptin	*Phyllomedusa bicolor*	S. America	62	[16]
1992	Brevinin	Rana brevipoda porsa	Asia	221	[17]
	Caerin	*Litoria splendida,*	Australia	29	[18]
1993	Esculentin	*Rana esculenta*	Europe	58	[19]
1994	Ranalexin	*Rana catesbeiana*	N. America	4	[20]
1995	Rugosin	*Rana rugosa*	Asia	8	[21]
1996	Temporin	Rana temporaria	Europe	119	[22]
	Uperin	*Litoria ewingi*	Australia	12	[23]
1998	Ranatuerin	Rana catesbeiana	N. America	51	[24]
	Maculatin	*Litoria eucnemis*	Australia	7	[25]
	Dermatoxin	*Agalychnis annae*	S. America	4	[26]
1999	Citropin	*Litoria citropa*	Australia	6	[27]
2000	Aurein	Litoria aurea	Australia	12	[28]
	Phylloxin	*Phyllomedusa bicolor*	S. America	2	[29]
	Palustrin	*Rana palustris*	N. America	27	[30]
	Kassinatuerin	*Kassina senegalensis*	Africa	5	[31]
2001	Tigerinin	Rana tigerina	Asia	4	[32]
	Pseudin	*Pseudis paradoxa*	S. America	4	[33]
	Distinctin	*Phyllomedusa distincta*	S. America	2	[34]
	Nigrocin	*Pelophylax nigromaculatus*	Asia	35	[35]
	Dahlein	*Litoria dahlii*	Australia	11	[36]
2002	Japonicin	*Rana japonica*	Asia	7	[37]
2003	Ranacyclin	*Rana esculenta*	Europe	13	[38]
2004	Ocellatin	Leptodactylus ocellatus	S. America	18	[39]
	Ascaphin	*Ascaphus truei*	N. America	13	[40]
2005	Phylloseptin	*Phylllomedusa hypochondrialis*	S. America	30	[41]
	Hylin	*Hyla biobeba*	S. America	3	[42]
2006	Lividin	*Odorrana livida*	Asia	5	[43]
	Pelophylaxin	*Pelophylax plancyi fukienensis*	Asia	6	[44]
2007	Dybowskin	*Rana dybowskii*	Asia	8	[45]
	Pleurain	*Rana pleuraden*	Asia	13	[46]
	Odorranain	*Odorrana graham*	Asia	31	[47]
	Hyposin	*Phyllomedusa hypochondrialis*	S. America	5	[48]
2008	Fallaxidin	*Litoria fallax*	Australia	3	[49]
2010	Nigroain	*Rana nigrovittata*	Asia	8	[50]
2011	Andersonin	*Odorrana andersonii*	Asia	6	[51]
	Cathelicidin	*Amolops loloensis*	Asia	9	[52]
2012	Hymenochirin	*Hymenochirus boettgeri*	Africa	16	[53]
2014	Frenatin	*Sphaenorhynchus lacteus*	S. America	6	[54]
2015	Defensin	*Theloderma kwangsiensis*	Asia	2	[55]

^1^ Based on the Antimicrobial Peptide Database (APD) (http://aps.unmc.edu/AP) in June 2020. The year indicates the discovery of the first member in the peptide family on an indicated continent. Some peptide families such as temporin and brevinin have been subsequently found on other continents. ^2^ This reference first reports the peptide in the family based on the APD.

**Table 2 antibiotics-09-00491-t002:** Amphibian antimicrobial peptides possess diverse activities ^1^.

Activity	APD Count	Examples
Antibacterial	982	Magainin 2, Brevinin-1, Temporin-SHf
AntiGram+	169	Temporin-1Ola, Aurein 5.2, Phylloseptin-S3
AntiGram−	84	XT-2, Dermaseptin DA4, Leptoglycin
Anti-MRSA	54	Ranalexin, Temporin-1GHd, Temporin-1OLa
Antifungal	490	Dermaseptin-S1, Temporin-1GHd, PGLa
Candidacidal	373	Maximin 1, Esculentin-1, Temporin A
Antivirus	45	Brevinin-1, Urumin, Temporin B,
Anti-HIV	36	Caerin 1.1., Maculentin 1.1, Dermaseptin-S1
Anti-parasitic	53	Dermaseptin-S1, Temporin L, Phylloseptin-S4
Anticancer	93	Magainin 2, Aurein 1.2, Dermaseptin
Anti-diabetes	15	Esculentin-1, Amolopin, Dermaseptin-B4
Anti-endotoxin	13	Temporin L, Cathelicidin-PP, Pseudin-2
Insecticidal	2	Escculentin-1, Magainin 2
Spermicidal	5	Magainin 2, Dermaseptin-S1, Dermaseptin-S4
Chemotactic	4	Temporin-A, Temproin B, Dermaseptin-S9
Wound healing	4	Magainin 2, Temporin A, Brevinin-2Ta
Antioxidant	19	Temproin-TP1, Pleurain-A1, Brevinin-1TP3
Protease inhibitor	4	Odorranain-B1, P2-Hp-1935
Anti-inflammatory	3	Cathelicidin-PY, Cathelicidin-PP, Esculentin-1GN

^1^ Accessed the APD (http://aps.unmc.edu/AP) on 22 July 2020.

**Table 3 antibiotics-09-00491-t003:** Different requirements of amphibian peptides for antibacterial and hemolytic properties.

Activity Spectrum ^1^	Count	Hydrophobic Content (Pho) ^2^	Net Charge	Lysine%
Hemolytic	163	55.6%	+2.67	12.6%
G+/G−	982	51.1%	+2.55	12.9%
G−	84	47.8%	+2.74	14.5%
G+	169	55.1%	+1.62	9.5%

^1^ Average hydrophobic contents and net charge obtained from the APD (http://aps.unmc.edu/AP) on 20 July 2020. G+/G−: antibacterial; G+: active primarily against Gram-positive bacteria; G−: active against primarily Gram-negative bacteria. Hemolytic: amphibian antimicrobial peptides with high hemolytic ability in the APD. ^2^ Pho is the total hydrophobic content calculated by summing those of isoleucine (I), valine (V), leucine (L), phenylalanine (F), cysteine (C), methionine (M), alanine (A), and tryptophan (W).

**Table 4 antibiotics-09-00491-t004:** Amino acid use in dermaseptins depends on peptide length ^1^.

Length Range	Peptide Count	Average Net Charge	L%	A%	G%	K%
21–25	17	+1.8	13.3	19.4	13.3	12.6
26–30	25	+2.6	12.8	21.0	11.5	13.4
31–35	20	+3.1	9.9	28.5	10.5	14.2

^1^ Obtained from the APD (http://aps.unmc.edu/AP) in June 2020.

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
