# Peer review of "Bioinformatic Analysis of 1000 Amphibian Antimicrobial Peptides Uncovers Multiple Length-Dependent Correlations for Peptide Design and Prediction"

_antibiotics, 2020, doi:10.3390/antibiotics9080491_

Round 1

Reviewer 1 Report

In this manuscript, the author summarized the development of amphibian antimicrobial peptides and analyzed their antimicrobial rules. The topic is interesting, and the manuscript is well organized. However, several comments should be addressed before accepting this manuscript for publication.

  1. The authors should introduce more information about the antimicrobial mechanisms of these peptides.

  1. What concentrations (nM or µM) are needed to use when killing bacterial? For example, the IC50 of these peptides.

  1. How to compare the antimicrobial capability of these peptides? Are they in the same molar concentration or weight concentration?

Author Response

In this manuscript, the author summarized the development of amphibian antimicrobial peptides and analyzed their antimicrobial rules. The topic is interesting, and the manuscript is well organized. However, several comments should be addressed before accepting this manuscript for publication.

  1. The authors should introduce more information about the antimicrobial mechanisms of these peptides.

Reply: Peptide mechanisms have been added to the Introduction section.

  1. What concentrations (nM or µM) are needed to use when killing bacterial? For example, the IC50 of these peptides.

Reply: The minimal inhibitory concentrations for membrane targeting peptides are observed in the micromolar range. However, some bacteriocins can inhibit bacterial growth at nanomolar with a defined non-membrane target. This information has been added to section 5 (Methods).

  1. How to compare the antimicrobial capability of these peptides? Are they in the same molar concentration or weight concentration?

Reply: The antimicrobial capability is reported in either micromolar or µg/ml in the literature. It is difficult to compare the data from different laboratories conducted under different conditions. To provide a useful positive data set, the antimicrobial peptide database does not register a peptide if its minimal inhibitory concentration (MIC) is greater than 100 µM or 100 µg/ml.

Reviewer 2 Report

In the present article, Guangshun Wang summarized the development of AMPs and analyzed some features of frog AMPs. Although the AMP database maintained by the author helps peptide-based drug design and translational medicine, this manuscript merely took on data from the database and finished a simple analysis of residue characteristics. It lacks comprehensive analysis in-depth to explore the origin and application of these peptide features. Additionally, the quality of the figures is too poor to be published. Thus, the academic significance of it for antibiotics is limited and I’m sorry to reject this manuscript.

Author Response

We have improved the manuscript substantially.  We have also improved the figures for publication.

Reviewer 3 Report

Major comments:

I suggest making a table with numbers of active AMPS against Gram-negative, Gram-positive bacteria etc. as explained in the text on page 3. The table will be a more meaningful way of summarizing the AMPs.  

For a complete overview of the very significant summary represented in this paper, it will be highly relevant to make an analysis where the structure of the frog peptides would be related to their specific biological activity so that one can deduce some information when designing AMPs with specific amino acid sequences.

Line 91 paragraph 2.3.

In addition to the summary of the number of antimicrobial activities of frog peptides, the author should summarize and add a section that will highlight the toxicity of frog peptides since for most of them there is available literature. Toxicity is a biological activity that maybe is not to the advantage of these molecules but it should not be neglected.

Line 109, Line 120- What is the mechanism of AMPs that show antiviral activities?

It will be nice to add a sentence for all of the activity which will point out the feature or that has been reported to be important for the observed activity.

Line 170- give the name of the database program used in addition to the reference.

Line 218 – why would these amino acids be useful for designing of AMPs? Would they be useful when designing specific AMPs or AMPS with broad-spectrum? Discuss briefly current efforts to synthesize synthetic or mimetic of natural frog peptides for therapeutic use and what is the success here?

Line-244 Here there is a discussion where the author refers to their own papers where this model has been used. However, this is limiting and studies that have used predictions or information of bioinformatic in silico analysis to design novel peptides should be briefly included in this section.

Line 252 – add the date of access to the link because the current database is constantly edited.

Line 270-correct for select-selected

Line 277- this part of the analysis belongs to the result section and not a discussion

Discussion:

The discussion part should include some more studies and perspectives of the application of the current understanding of the structural requirements for active AMPs.

Line 282-283- using questions in the conclusion could be avoided and merged with the answer.

>Line 447- Please give reference to the data that has been used to generate Table 2.

Line 458 – delete the extra comma.

Comments: Figure legends

All the figure legends should be a stand lone explanatory. Reference to text, e.g. see the text, should be deleted.

Figure 3. North America's abbreviation in the figure should be corrected. Same in Figure 4.

Author Response

I suggest making a table with numbers of active AMPS against Gram-negative, Gram-positive bacteria etc. as explained in the text on page 3. The table will be a more meaningful way of summarizing the AMPs.  

Reply: This is a good suggestion. We have added a new Table 2 that summarizes other activities of amphibian AMPs.

For a complete overview of the very significant summary represented in this paper, it will be highly relevant to make an analysis where the structure of the frog peptides would be related to their specific biological activity so that one can deduce some information when designing AMPs with specific amino acid sequences.

Reply:  We have discussed the importance of amphipathic structure, aromatic phenylalanine as a membrane anchor (see new Fig. 7), the plasticity of the helical backbone (from sidechain changes via residue deployment to backbone changes), and the synthesis of mimics. 

Line 91 paragraph 2.3.

In addition to the summary of the number of antimicrobial activities of frog peptides, the author should summarize and add a section that will highlight the toxicity of frog peptides since for most of them there is available literature. Toxicity is a biological activity that maybe is not to the advantage of these molecules but it should not be neglected.

Reply: We have added a new section 2.4 on “Toxicity of amphibian peptides”. 

Line 109, Line 120- What is the mechanism of AMPs that show antiviral activities?

Reply: We have consolidated the section by describing different viral targets for certain amphibian AMPs.

It will be nice to add a sentence for all of the activity which will point out the feature or that has been reported to be important for the observed activity.

Reply: This is a good suggestion that makes each section more interesting to read.

Line 170- give the name of the database program used in addition to the reference.

Reply: We have clarified this sentence as below:

“…by using the database statistical analysis after each peptide search”.

Line 218 – why would these amino acids be useful for designing of AMPs? Would they be useful when designing specific AMPs or AMPS with broad-spectrum? Discuss briefly current efforts to synthesize synthetic or mimetic of natural frog peptides for therapeutic use and what is the success here?

Reply: The frequently occurring amino acids represent both hydrophobic and charged components of AMPs, making them useful for peptide design as we have already demonstrated (ref. 39). We also point out that the ratio between basic and hydrophobic amino acids determines peptide activity spectrum. Based on this useful suggestion, we have expanded the discussion by covering peptide length, charge, hydrophobic content, and post-translational modification. This is followed by discussion of peptide design as well as peptide mimicries. 

Line-244 Here there is a discussion where the author refers to their own papers where this model has been used. However, this is limiting and studies that have used predictions or information of bioinformatic in silico analysis to design novel peptides should be briefly included in this section.

Reply: We have discussed additional examples from the literature.

Line 252 – add the date of access to the link because the current database is constantly edited.

Line 270-correct for select-selected

Reply: Corrected.

Line 277- this part of the analysis belongs to the result section and not a discussion

Discussion:

Reply: We have made it clearer by adding “Methods” to the title of section 4. For clarity, subtitles were also inserted.

The discussion part should include some more studies and perspectives of the application of the current understanding of the structural requirements for active AMPs.

Reply: We have incorporated structural information into our discussion by pointing out (1) the role of proline, (2) the peptide backbone structure, and (3) the significance of aromatic phenylalanine as a membrane anchor.

Line 282-283- using questions in the conclusion could be avoided and merged with the answer.

Reply: This is a good suggestion. We have removed such questions in the Conclusion.

>Line 447- Please give reference to the data that has been used to generate Table 2.

Reply: We have added a footnote that indicates the source of the data. In addition, we also provided references for Table 1.

Line 458 – delete the extra comma.

Reply: Deleted. 

Comments: Figure legends

All the figure legends should be a stand lone explanatory. Reference to text, e.g. see the text, should be deleted.

Reply: fixed. 

Figure 3. North America's abbreviation in the figure should be corrected. Same in Figure 4.

Reply: fixed. 

Round 2

Reviewer 2 Report

In the revised article, Guangshun Wang significantly increased the quality of the manuscript. The comprehensive summary of AMPs provides insights into drug design and treatment development. Abundant and accurate references also greatly benefit the application of AMPs. Meanwhile, the quality of the figures has been improved a lot, which meets the requirement of publication. Thus, I’m pleased to accept this manuscript on Antibiotics.